# Quantification of Iron Release from Native Ferritin and Magnetoferritin Induced by Vitamins B_2_ and C

**DOI:** 10.3390/ijms21176332

**Published:** 2020-08-31

**Authors:** Oliver Strbak, Lucia Balejcikova, Martina Kmetova, Jan Gombos, Alzbeta Trancikova, Michal Pokusa, Peter Kopcansky

**Affiliations:** 1Biomedical Center Martin, Jessenius Faculty of Medicine in Martin, Comenius University in Bratislava, Mala Hora 4, 036 01 Martin, Slovakia; alzbeta.trancikova@uniba.sk (A.T.); michal.pokusa@uniba.sk (M.P.); 2Institute of Hydrology, Slovak Academy of Sciences, Dubravska cesta 9, 841 04 Bratislava, Slovakia; balejcikova@uh.savba.sk; 3Department of Medical Biochemistry, Jessenius Faculty of Medicine, Comenius University in Bratislava, Mala Hora 4, 036 01 Martin, Slovakia; martinamihalikova09@gmail.com (M.K.); gomboss.jan@gmail.com (J.G.); 4Institute of Experimental Physics, Slovak Academy of Sciences, Watsonova 47, 040 01 Kosice, Slovakia; kopcan@saske.sk

**Keywords:** ferritin, magnetoferritin, mineral core, reduction, ascorbic acid, riboflavin, ferrous ions, time dependence, toxicity

## Abstract

Various pathological processes in humans are associated with biogenic iron accumulation and the mineralization of iron oxide nanoparticles, especially magnetite. Ferritin has been proposed as a precursor to pathological magnetite mineralization. This study quantifies spectroscopically the release of ferrous ions from native ferritin and magnetoferritin as a model system for pathological ferritin in the presence of potent natural reducing agents (vitamins C and B_2_) over time. Ferrous cations are required for the transformation of ferrihydrite (physiological) into a magnetite (pathological) mineral core and are considered toxic at elevated levels. The study shows a significant difference in the reduction and iron release from native ferritin compared to magnetoferritin for both vitamins. The amount of reduced iron formed from a magnetoferritin mineral core is two to five times higher than from native ferritin. Surprisingly, increasing the concentration of the reducing agent affects only iron release from native ferritin. Magnetoferritin cores with different loading factors seem to be insensitive to different concentrations of vitamins. An alternative hypothesis of human tissue magnetite mineralization and the process of iron-induced pathology is proposed. The results could contribute to evidence of the molecular mechanisms of various iron-related pathologies, including neurodegeneration.

## 1. Introduction

Iron played a role in the origin of life on Earth [1] and, except for Lactobacillus and certain strains of Bacillus [2], it is an essential element for living organisms [3]. It is involved in many fundamental metabolic processes such as photosynthesis, respiration, DNA synthesis, cell proliferation, and differentiation. The vast majority of biological iron is bound to metalloproteins, with only minor amounts of free or loosely bound iron forming a “labile iron pool”, as although iron is an essential element for the survival of cells, excess free iron is extremely toxic for organisms [4]. In humans, elevated levels of iron are associated with various pathological processes including neuroinflammation [5], neurodegeneration [6], cardiovascular disease [7], and cancer [8], but it is not yet known whether iron accumulation is the cause or the consequence of the pathology. However, it is generally accepted that the toxicity of iron results from the ability of ferrous ions to produce hydroxyl radicals through the Fenton and Haber–Weis reactions [9]. Fe^2+^ cations immediately oxidize biological molecules, including nucleic acids, proteins, and lipids, impairing their function [10]. Free ferrous ions produce free radicals that increase oxidative stress and can cause disease [11]. To prevent oxidative damage, a balanced ratio of free radicals and antioxidants is necessary to achieve equilibrium [12], and also functional ferritin, which is crucial for iron homeostasis [13]. Moreover, soluble Fe^2+^ ions react instantly with oxygen, forming insoluble iron oxide precipitates at neutral and alkaline pH [14]. It is widely accepted that disrupted iron homeostasis results in iron accumulation [6,15] and the mineralization of nanosized iron oxide particles, in particular as magnetite [16].

Despite the discovery of biogenic magnetite nanoparticles in human brain tissue in 1992 by Kirschvink et al. [17], the mechanism of its formation and role in humans is still unknown, although it is well described, for example, in magnetotactic bacteria, where it enables navigation in the geomagnetic field [18]. While the reason for the presence of magnetite in human tissue is not known, it has been proposed that a precursor of iron accumulation and magnetite mineralization is ferritin [19,20], an intracellular particle containing a protein envelope (12 nm) with a central cavity (8 nm) and an inner mineral core (2–7 nm) [21]. The primary role of ferritin is iron storage (in the non-toxic ferric state); for further biological use its reduction is required. In physiological conditions, the ferrous ions can quickly diffuse to the ferritin interior through the eight narrow hydrophilic threefold channels [22]. During the transfer, Fe^2+^ cations are oxidized to ferric ions, forming a ferrihydrite-like mineral core in the ferritin’s cavity. Oxidation of iron occurs in the H-subunit of the ferritin particle, while the L-subunits assist iron nucleation and mineralization [23].

In 2004, Quintana et al. utilized electron nanodiffusion and electron microscopy to show that the ferritin mineral nucleus of patients with Alzheimer’s disease, referred to as “pathological” ferritin, differs structurally from native “physiological” ferritin [24]. In physiological ferritin, the mineral core consists mainly of hexagonal ferrihydrite, hematite, and a smaller phase of magnetite. In contrast, the core of pathological ferritin consists primarily of cubic structures, such as magnetite and wustite, and to a lesser extent ferrihydrite, but with no hematite. These conclusions have been confirmed in a recent study [25], where muon spin rotation was utilized to demonstrate that ferritin particles have a crystalline phase with large magnetocrystalline anisotropy that is compatible with magnetite or maghemite. The altered composition of the ferritin’s mineral core could be associated with pathology. In the following study, magnetoferritin (MF) is used as a model system of pathological ferritin, as both are composed of apoferritin and magnetite nanoparticles [26]. The altered composition of the pathological ferritin mineral core prompts a new question of fundamental importance: which biochemical processes enable the transformation of the physiological ferrihydrite-like mineral core into the pathological magnetite-like mineral core under physiological conditions? Answering this may lead to an understanding of the pathological processes associated with iron accumulation and mineralization, and lead to potential therapeutic opportunities.

As the ferrihydrite mineral in native ferritin (NF) consists solely of ferric ions, and magnetite in pathological ferritin is composed of both ferric and ferrous ions, the reduction mechanism must be determined. The most potent reducing agents of intact ferritin include flavins [27,28] and ascorbate [29,30]. These single-electron reducing agents are capable of reducing the ferrihydrite mineral core, as well as releasing iron from the ferritin [31,32]. Therefore, to find a possible transformation mechanism, this study focused on vitamin C (ascorbic acid) and vitamin B2 (riboflavin), which are biologically essential molecules capable of reducing ferritin’s mineral core; they are also present in relatively high concentrations in cells [33,34]. The aim was to determine whether both molecules are able to reduce the mineral core of ferritin/MF and release the reduced iron from the particle. It has been shown that any ferrous ions released are quickly reoxidized by ferritin molecules, and their concentration is difficult to determine due to its low value [35,36]. In vitro studies have characterized the ferritin rate constants as follows; 1200 mol Fe/mol ferritin/second for the formation of the mineral core [37], and 0.52 mol Fe/mol ferritin/second for release from the protein [38]. Based on this, the time dependence of the rate of production of ferrous ions, based on the interaction of different concentration levels of both vitamins with native (physiological) ferritin, as well as in MF (as a model system for pathological ferritin), was determined.

## 2. Results

### 2.1. Magnetoferritin Characterization

Controlled in vitro synthesis was used to prepare MF with three different iron loading factors (LF) (calculated using Equation (1)): LF^MF1^ = 553, LF^MF2^ = 733, and LF^MF3^ = 873, with an error of less than 2%. The LF of NF in colloidal solution is 868 (Table 1).

The average hydrodynamic diameter (<D_HYDR>_) of the different samples, obtained by dynamic light scattering (DLS) due to the Brownian motion of the nanoparticles in colloidal solution, is shown in Table 1. Nanoparticles possess an electrical bilayer that is dependent on their concentration, and the pH and ionic strength of the surrounding medium; thus, the <D_HYDR_> of ferritin particles is greater than native apoferritin biomacromolecules (10–12 nm). A higher <D_HYDR>_ signifies the presence of larger objects, due to protein interactions or magnetic attraction between MF nanoparticles; the presence of single particles in the solution is also possible.

Zeta (ζ) potential measurements confirmed a total negative charge for all prepared samples (Table 1). The absolute values of the zeta potential signify good colloidal stability of the samples, which means good resistance to sedimentation.

The dependences of the magnetization on the intensity of the applied magnetic field was measured by superconducting quantum interference device (SQUID) magnetometry at 290 K, which corresponds to a temperature at which no hysteresis was observed (Figure 1). The decreasing trend that deforms the magnetization curves indicates the presence of a phase with diamagnetic properties. This behavior was probably because the samples were in a liquid state, and the diamagnetic contribution may belong to the water, the contribution of which was not subtracted. The observed behavior may also be due to the presence of another diamagnetic organic component in the sample—the protein.

Based on recent work in this laboratory, it is proposed that the mineral core of the MF samples in this study does not consist of only one phase but is a polyphase [39]. In 2017, unexpected nonlinear behavior of magnetization field dependencies with LF growth was observed for the first time [40]. It can be assumed that in the case of higher LFs, the preferable formation of FeO(OH)-like compounds can affect the magnetization value. According to Wong et al. [41], the basic equation for the synthesis of magnetite in an apoferritin shell is:

3 Fe^2+^ + 2 (CH_3_)_3_NO + 4 H_2_O → Fe_3_O_4_ + 2 (CH)_3_N + 6 H^+^ + H_2_O_2_(1)

During this process, protons are formed, which could change the solution pH and affect oxidation and hydrolysis, followed by γ-FeO(OH) (lepidocrocite) formation [39]. Under these conditions, γ-FeO(OH) can be transformed into γ-Fe_2_O_3_, so the inorganic core of MF may contain more than two phases.

### 2.2. Mineral Core Reduction

To understand the potentially toxic and initiating effect of iron in the pathological processes associated with biogenic iron accumulation, the rate of release of ferrous ions from ferritin must be determined because this process has been proposed as a precursor of pathological iron mineralization [19,20]. A comparison of the amount of iron released from the cores of NF and MF as a model system of pathological ferritin could indicate whether a transformed magnetite-containing core has the potential to initiate oxidative stress (disease) or might only be a result of pathological processes. In this case, the ferrous ions are the most likely candidate for the toxic effects on cells (due to the Fenton and Haber–Weiss reaction [9]) and pathology initiation. Here, the so-called “pathological” ferritin is characterized by a change in the composition of the mineral core, with ferrihydrite being transformed into magnetite [24]. As magnetite contains both ferrous and ferric ions, and ferrihydrite involves only ferric ions, the reduction mechanism is required. The most powerful naturally occurring reducing agents of intact ferritin are flavins [27,28] and ascorbate [29,30].

Figure 2 and Figure 3 and Appendix A show the time dependences of ferrous ion release from NF and MF, as a model system of pathological ferritin, in the presence of ascorbic acid (vitamin C—Figure 2 and Appendix A) and riboflavin (vitamin B_2_—Figure 3 and Appendix A). In both cases, the release of ferrous ions from the mineral core of NF as well as MF is observed. These results indicate successful reduction by both vitamins and the release of ferrous ions from the ferritin’s core regardless of its iron mineral phase. The concentrations of both vitamins were selected to mimic the approximate physiological concentration in human cells (healthy erythrocytes): vitamin B_2_ ≈ 140 µM [42] and vitamin C up to 60 µM [43] (approximated to a concentration of 100 µM for both vitamins B_2_ and C), including slightly lower (10 µM vitamins B_2_ and C) and slightly higher (1000 µM vitamin B2 and 500 µM vitamin C) concentrations than physiological ones. In reality, the actual level in cells varies with the instantaneous requirements of the organism and food intake.

For vitamin C the time dependence of ferrous ion release appears more or less random (Figure 2 and Appendix A), in contrast, for vitamin B_2_ the ferrous ion concentration levels continuously increase after approximately 5 h (Figure 3 and Appendix A). This is also reflected in the increase of the correlation coefficient values in the case of vitamin B_2_, as shown in Appendix A. All values of the correlation coefficients are positive, and for most of the compared pairs, they are higher than 0.5 for vitamin C and 0.7 for vitamin B_2_, which points to a relatively strong relationship.

For NF, the range of iron release rates from the mineral core is only slightly influenced by the vitamin concentration, especially for vitamin C (Figure 4 and Figure 5). With increasing vitamin C concentration, the number of reduced iron ions also increases. In contrast, for MF, it seems that the level of the vitamin does not affect the amount of released Fe^2+^ cations significantly. Compared to NF, MF is characterized by a steady state, independent of the vitamin concentration. This points to a threshold concentration in the case of the magnetite core, above which the amount of reduced ferric iron is in balance with the oxidation of free ferrous iron by a protein envelope. However, the amount of “free” iron in the case of the MF core is approximately two to five times greater than in the case of the NF core (Figure 4 and Figure 5). This is a key result concerning pathological action. The number of ferrous ions released from NF nuclei is always fewer than the number of ferrous ions liberated from MF nuclei, regardless of the LF and reducing agent concentration (vitamin C—Figure 2 and Figure 4; vitamin B_2_—Figure 3 and Figure 5). This difference is more evident in the case of ascorbic acid. Surprisingly, the LF of MF has no significant effect on the rate of mineral core reduction and the liberation of Fe^2+^ cations over time for both vitamins, as evidenced by the median value in the box plots for each LF of MF samples. The increased concentration level of released iron from the magnetite-like mineral core (MF) compared to the ferrihydrite-like mineral core (NF) could be a crucial factor in the initiation of various pathological processes that are associated with iron accumulation. The consequences of this are dealt with in more detail in the discussion section.

For a complementary quantitative comparison, the 95% confidence interval (mean ± 2 SEM (standard error of the mean)) of all samples per hour is shown in Table 2. In Figure 6 and Figure 7, the interval is also shown graphically for both vitamins. For better visualization, separate graphs showing the 95% confidence interval for each vitamin concentration and LF individually are included in Appendix A. These results confirm the previous findings (Figure 2, Figure 3, Figure 4 and Figure 5) that the “production” of “free” iron in the case of the MF core is approximately two to five times greater than in the case of the NF core.

Qualitatively interesting information is provided by the MF/NF ratio in Figure 8 and Figure 9. This ratio shows the difference in the dynamics of the mineral core reduction induced by vitamins C and B_2_, respectively. While Figure 8 points to stable dynamics (small variance of values, from 1.5 to 4), in the case of vitamin B_2_ (Figure 9), the range of iron release is much wider (from 2 up to approximately 12). However, in both cases, a decrease in the median value of the MF/NF ratio can be seen with increased vitamin concentration. This points to the same mechanism of mineral core reduction by both vitamins as the iron reoxidation by a protein envelope should be the same in both cases. In the case of vitamin C, it is evident that there is a steady-state mechanism of mineral core reduction, independent of its LF (Figure 8). There is a more turbulent situation in the case of vitamin B_2_, especially for the MF2 sample and 100 µM riboflavin (Figure 9). Unexpectedly, the mineral core with a lower LF (MF2) produces more iron than the mineral core with a higher LF (MF3) (Figure 9). As this trend is more or less the same at all vitamin B_2_ concentrations, it can be concluded that the synthesis of the MF2 core results in a different mineral phase composition that is prone to reduction. It may inadvertently contain a higher magnetite ratio compared to the MF3 mineral core or another phase that has already been discussed [40]. Alternatively, it could be caused by the fact that a higher LF could contain γ-FeO(OH) (lepidocrocite), which is more similar to ferrihydrite [39]. However, this fact should not affect our conclusions, which will be discussed in more detail later.

## 3. Discussion

As previously described, the mineral core of physiological ferritin consists of ferrihydrite, while the mineral core of ferritin associated with pathology consists of magnetite [24,25]. The size of the magnetite nanocrystals ranges from 10 to 70 nm [17]. The results above clearly show that both vitamins (B_2_ and C) are capable of reducing the mineral core of NF as well as MF, as a model system of pathological ferritin. The release of ferrous ions was sufficient to be detected, unlike the findings of previous studies, where the released ferrous ions were quickly reoxidized by ferritin molecules making their detection impossible [35,36]. In vitro studies characterized the ferritin rate constant for the release of iron as 0.52 mol Fe/mol ferritin/second [38]. Riboflavin and ascorbic acid have increased this value from 10.3 to 35.3 mol Fe/mol NF/second (Table 2, first row for NF). Although it exceeds the in vitro value for iron release, it does not reach the rate constant required for the formation of the mineral core in vitro (1200 mol Fe/mol ferritin/second) [38]. In MF, the concentration of released (free) iron is in the order of tens to hundreds of ng/mL (Table 2).

Comparison of the average values of free iron released after reduction from NF, as shown in Table 2, with the total iron concentration in our NF samples, which is 5200 ng/mL, indicates that theoretically the reduction rate induced by vitamins over 12 h is not sufficient to provide enough ferrous ions to transform the ferrihydrite mineral core into magnetite. Magnetite consists of approximately 69% Fe_2_O_3_ (Fe^3+^) and 31% FeO (Fe^2+^). Thus, for the complete transformation of a ferrihydrite core to magnetite, it would be necessary to oxidize about a third of the iron atoms, approximately 1700 ng/mL. The average values of released iron per hour from NF are in the range of 17.7 to 68.6 ng/mL (Table 2). This suggests that the reduction mechanism induced by vitamins alone would not be sufficient to initiate ferrihydrite-to-magnetite transformation to the extent found in human brain tissue (1–7 µg/g of dry tissue [16]). Therefore, another mechanism is required, which will be discussed in more detail in Section 3.1.

In general, the results of this study can be summarized in three points:Biologically essential vitamins (B_2_ and C) are capable of ferritin mineral core reduction at a rate that exceeds the rate of iron release from “untouched” ferritin by 20 to 68 times per second. From the long-term point of view, this can result in the accumulation of toxic iron (ferrous ions) in cells, giving rise to a higher risk of oxidative stress.An increase in the concentration of the reducing agent increases iron release only from the NF mineral core. The MF mineral core seems to be unaffected by different concentrations of vitamins (Figure 4 and Figure 5).The amount of reduced iron produced in the MF mineral core is two to five times higher than in the NF mineral core (Figure 2 and Figure 3).

These three findings lead to the conclusion that the magnetite core in pathological ferritin has the potential to increase oxidative stress in cells, which may cause pathology; this process is caused by the presence of the naturally occurring biologically essential molecules—vitamins B_2_ and C. The first point indicates the outcome that the reduction rates of the NF and MF cores in the presence of both vitamins are large enough for the accumulation of toxic ferrous ions, which are capable of oxidative stress. The second point indicates that an increase in vitamin concentration increases iron release from the NF mineral core and may indicate that there is an increased threat of oxidative stress purely from an increase in vitamin intake. This agrees with the finding that increased vitamin C intake from supplements is associated with an increased risk of cardiovascular disease [44]. The third, and probably most important result, points to the fact that ferrous iron release from the mineral core is two to five times greater in the case of MF compared to NF. This indicates that with age, the risk of iron-induced oxidative stress increases several times (as discussed in more detail below). Increased oxidative stress can subsequently lead to the emergence of various pathological processes, including neurodegenerative disorders and cancer [45], but in vivo studies are needed to support these in vitro findings.

### 3.1. A New Proposal for the Origin of Magnetite in Human Tissue and the Initiation of Associated Pathology

The results presented above suggest a new mechanism for the emergence of iron-induced pathology. They can also be associated with a newly proposed alternative explanation for iron accumulation and magnetite mineralization in human tissue, which has a strong historical context. In the past, due to the emergence of cyanobacteria and the evolution of oxygen-based photosynthesis, free (biologically usable) iron became a scarce commodity [46]. As a result, biological systems, including currently living organisms (and humans), have not developed a specific and regulated excretion mechanism for iron. In humans, it is generally accepted that iron levels are controlled exclusively by intestinal iron absorption [47]. Iron excretion occurs only at a basal rate, regardless of iron absorption, through uncontrolled processes including blood loss, intestinal epithelium turnover, and loss of old skin [48]. Therefore, excess iron must be stored somewhere inside the tissue.

It is well documented that with increased age, iron concentration levels increase in biological systems [49], including humans [6], where iron is stored exclusively in ferritin or hemosiderin molecules. As the intracellular iron deposition pathway is defined in the order of (1) labile pool, (2) ferritin, and (3) hemosiderin [50], it can be considered that ferritin is the primary means of excess iron storage, due to the lack of a regulated excretion mechanism. This is also consistent with previous proposals, which assume ferritin is a precursor of “pathological” iron accumulation [19,20]. In addition, no other mechanism is currently known, apart from that in ferritin, that allows the mineralization of iron oxides in the human body. As the magnetite in the pathological ferritin has 1.3 times the density of ferrihydrite in physiological ferritin (5.175 g/cm^3^ vs. 3.96 g/cm^3^), it can be supposed that ferrihydrite-to-magnetite transformation is an effective and natural tool for storing excess iron in tissue. Magnetite mineralization would thus be a substitute for a non-existent iron excretion regulatory mechanism. As a result, with the aging of the organism, an increase in the concentration of magnetite should be observed, which is consistent with previous findings [6].

Hypothetically, aging would thus be a critical factor in the development of pathology through the increased production of free radicals, as a consequence of elevated ferrous iron generation because of accumulated magnetite. Although the concentrations of free iron released, whether from the NF or MF mineral core (tens to hundreds ng/mL), do not exceed the physiological values of “labile cell iron”, which is in the range of 0.5 to 1.5 µM [51], the long-term excess of ferrous ions could disturb the fine balance of the cell ROS-antioxidants and slowly contribute to the initiation of pathology that would culminate in advanced age, as is typical, for example, in neurodegenerative disorders [52]. From this point of view, previous studies that have shown increased magnetite concentrations in the brains of patients with neurodegenerative disorders [16,17] may simply point to the fact that neurodegeneration has developed in subjects with elevated levels of magnetite in the tissue, whether due to age, diet, environmental [53], or genetic factors.

However, at present, little is known about the role of magnetite in the physiology and pathology of the human body. There is even a recent hypothesis that connects magnetite with memory storage in neurons [54]. To better understand its role in tissue and the emergence of various pathological processes requires further research, including other biochemical compounds and in vivo studies. The fundamental question regarding physiological magnetite mineralization remains unanswered, as it is has been shown that in the laboratory magnetite formation requires an alkaline environment and temperatures higher than those observed physiologically [40].

## 4. Materials and Methods

### 4.1. Chemicals

Ammonium ferrous sulfate hexahydrate ((NH_4_)_2_Fe(SO_4_)_2_.6H_2_O), horse spleen apoferritin in 0.15 M NaCl, ethanol (C_2_H_6_O), horse spleen ferritin in 0.15 M NaCl, hydrogen peroxide (H_2_O_2_), 3-[(1,1-dimethyl-2-hydroxyethyl)amino]-2-hydroxypropanesulfonic acid (AMPSO), sodium hydroxide (NaOH), trimethylamine N-oxide (Me_3_NO), L-ascorbic acid (C_6_H_8_O_6_), riboflavin (C_17_H_20_N_4_O_6_), ferrozine, and 4-(2-hydroxyethyl)-1-piperazineethanesulfonic acid (HEPES) were obtained from SIGMA-Aldrich (Saint-Louis, MO, USA); Coomassie brilliant blue G 250 was obtained from Fluka (Buchs, Switzerland); hydrochloric acid (HCl) from ITES (Vranov nad Toplou, Slovakia); potassium thiocyanate (KSCN) from Slavus (Bratislava, Slovakia); phosphoric acid (H_3_PO_4_) and ethanol (C_2_H_6_O) from Centralchem (Bratislava, Slovakia); and distilled water.

### 4.2. Synthesis of Magnetoferritin

MF was prepared by controlled in vitro thermooxidation. Firstly, a 0.05 M AMPSO buffer was prepared and the pH was adjusted to 8.6 using 2 M NaOH solution, utilizing a pH meter (Mettler Toledo SevenEasy^TM^ S20-KS, Greifensee, Switzerland) and pH electrode (Mettler Toledo Inlab^®^ Science Pro). The stock solution was deaerated using nitrogen for 1 h and used as a dispersion medium to apply the required amount of native apoferritin. The reaction bottle with an apoferritin solution was hermetically enclosed and placed in a water bath. The temperature was set to 65 °C and kept constant throughout the synthesis via a magnetic stirrer, with heating (IKA C-MAG HS 7, Staufen, Germany). As a ferrous ion source, a 0.1 M deaerated solution of Mohr’s salt ((NH_4_)_2_Fe(SO_4_)_2_.6H_2_O) was prepared, and as an oxidant source, a stoichiometric amount of a 0.07 M deaerated solution of trimethylamine N-oxide was prepared. The gradual addition of reactants to the reaction bottle was performed in 10 steps over 100 min using syringes. Three types of MF samples with different LFs, representing the average number of iron atoms per one apoferritin biomacromolecule, were prepared for physico-chemical characterization and interaction studies.

### 4.3. Quantitative Determination of the Loading Factor

Quantitative analysis of the LF was performed using an ultraviolet–visible (UV–VIS) spectrophotometer (SPECORD 40, Analytik Jena, Jena, Germany) at 25 °C with a precision of approximately 1%. The mass concentration of iron atoms $c_{m}^{Fe}$ was determined after oxidation of Fe^2+^ to Fe^3+^ ions with 3% H_2_O_2_ in acidic conditions (35 % HCl) at 50 °C for 30 min. The addition of 1 M KSCN resulted in the formation of the red thiocyanate complex, Fe[Fe(SCN)_6_], and its absorbance was measured using a wavelength of 450 nm. From the calibration curve, using the regression equation, the corresponding mass concentration of iron atoms was calculated. The standard Bradford method was used for the determination of the mass concentration of NA, $c_{m}^{NA}$ [55]. The absorbance of the blue-colored complex of the Bradford agent with amino acid residues from proteins was detected at a wavelength of 595 nm, after 5 min incubation at 25 °C. From the calculated ratio of $c_{m}^{Fe}$ and $c_{m}^{NA}$ in a given volume of sample using the known molecular weights of apoferritin and iron, respectively, the loading of MF was calculated according to the Equation (2):(2)$$LF=\frac{c_{m}^{Fe}.\;M_{NA}}{c_{m}^{NA}.\;M_{Fe}}$$

### 4.4. Measurement of Hydrodynamic Diameters

The hydrodynamic diameter of the colloidal MF solution was measured using a Zetasizer NanoZS 3600 (Malvern Instruments, Malvern, UK). This instrument uses the principle of DLS, also known as photon correlation spectroscopy or quasi-elastic light scattering. This method allows the analysis of the intensity of the fluctuation of scattered light from particles in solution, which are undergoing Brownian motion. Measurement of their rate of diffusion in a liquid medium is related to their size, according to the Stokes–Einstein equation. The average hydrodynamic diameter <D_HYDR_> was measured using disposable polystyrene cuvettes at 25 °C. The size distribution was plotted using Zetasizer software as a function of the relative number of particles to their size, with the hydrodynamic diameter representing the maximum of the curve. The measurement of each sample’s hydrodynamic size was performed three times; the resulting <D_HYDR_> was the average of these values.

### 4.5. Zeta Potential Determination

Zeta potential is the electrokinetic potential in colloidal solutions, which is formed between the electrical bilayer of the particles and the surrounding medium in which they are dispersed. The zeta potential, as the main parameter of colloidal stability, was measured in capillary cells using a Zetasizer NanoZS 3600 (Malvern Instruments) using the principle of Laser Doppler velocimetry combined with electrophoresis.

### 4.6. SQUID Magnetometry

The magnetic properties of the prepared samples in a liquid medium were measured using a SQUID magnetometer MPSM 5XL (Quantum Design, San Diego, CA, USA). The hysteresis loops were measured at a temperature of 290 K in the range of the induced magnetic field up to 3 T.

### 4.7. The Study of Ferritin and Its Derivates’ Interaction with Ascorbic Acid and Riboflavin

The ferrozine assay was used for the quantitative determination of free iron ions after interaction with ascorbic acid and riboflavin at selected time intervals for 12 h. A solution of 0.02 M HEPES, adjusted using 2M NaOH to the physiological pH of 7.42, was used in calibrations and as a reaction medium. After the interaction of ferritin and MF with ascorbic acid and riboflavin, ferrozine was added, and the absorbance of the purple-colored complex of ferrozine-Fe(II) was measured after 1 min of incubation at a wavelength of 562 nm with an error of less than 1%. Each mixture was measured ten times, and the absorbance was averaged. Each reaction was performed at a temperature of 25 °C and protection against oxygen was facilitated by closing an Eppendorf tube.

### 4.8. Data Processing and Statistics

The software tool Matlab 2019a (Mathworks Inc., Natic, MA, USA) was used for data processing and statistical calculations. Ferrous ion concentrations, after release from the protein envelope, are reported as the mean ± standard deviation (SD) (line plots) and as the median with interquartile range (IQR) (box plots) of twelve consecutive measurements per hour (Figure 2, Figure 3, Figure 4 and Figure 5; Appendix A). For quantitative analysis, the 95% confidence interval (mean of the means per hour ± SEM) of the free ferrous ion concentration after reduction from the NF and MF mineral core per hour was used (Figure 6 and Figure 7; Appendix A). Correlation coefficients were calculated using a predefined function in Matlab (Appendix A).

## 5. Conclusions

At present, biogenic iron is attracting increasing attention as it is linked to several pathological processes, including neurodegeneration and cancer. One of the characteristic features of these pathological processes is the accumulation of iron in the form of nanoscale iron oxide particles. However, it is still unclear whether this accumulation is due to pathology or is a trigger for it. The results in this paper show that accumulated iron in the form of magnetite in the presence of biologically required molecules (vitamins B_2_ and C) has the potential, several times higher than normal, to produce ferrous ions which are a risk factor for oxidative stress and could initiate pathological processes, including neurodegeneration and cancer. This leads to an alternative, still unpublished, hypothesis regarding the existence of magnetite in human tissue and a higher risk of disease developing in old age. These results may help to reveal the background to the molecular mechanism of various iron-related pathologies, including neurodegeneration and cancer.

## Figures and Tables

**Figure 1 ijms-21-06332-f001:**
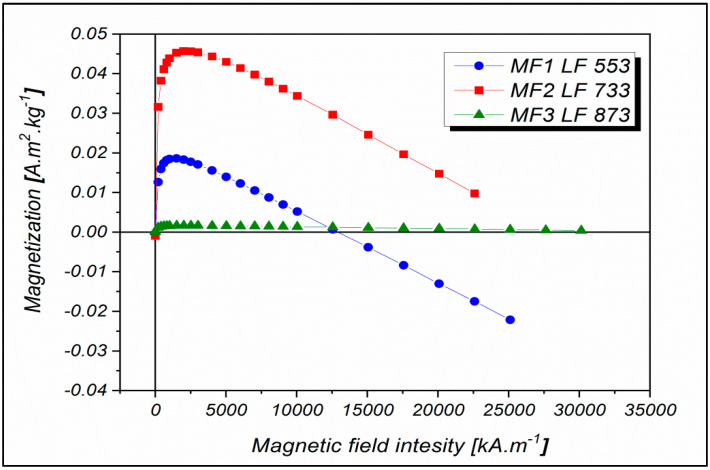
Magnetization curves measured at 290 K in a magnetic field with intensity up to 30,000 kA m^−1^ for MF with LFs equal to 553, 733, and 873.

**Figure 2 ijms-21-06332-f002:**
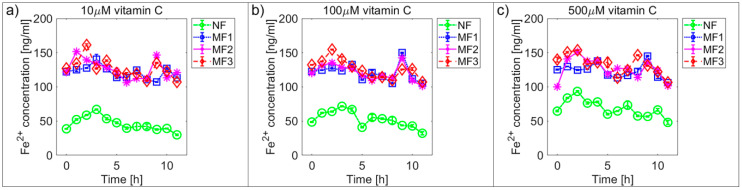
Comparison of the time dependence of ferrous ion release (mean ± SD) from NF and MF with various LFs, induced by different concentrations of ascorbic acid (vitamin C) as a reducing agent. (**a**) Vitamin C concentration = 10 µM; (**b**) vitamin C concentration = 100 µM; (**c**) vitamin C concentration = 500 µM.

**Figure 3 ijms-21-06332-f003:**
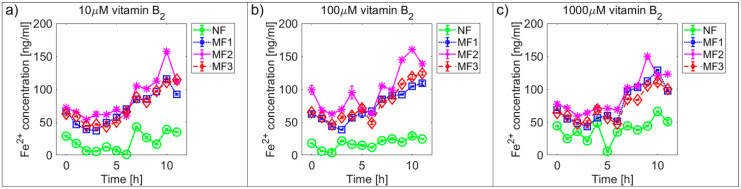
Comparison of the time dependence of ferrous ion release (mean ± SD) from NF and MF with various LFs, induced by different concentrations of riboflavin (vitamin B_2_) as a reducing agent. (**a**) Vitamin B_2_ concentration = 10 µM; (**b**) vitamin B_2_ concentration = 100 µM; (**c**) vitamin B_2_ concentration = 1000 µM.

**Figure 4 ijms-21-06332-f004:**
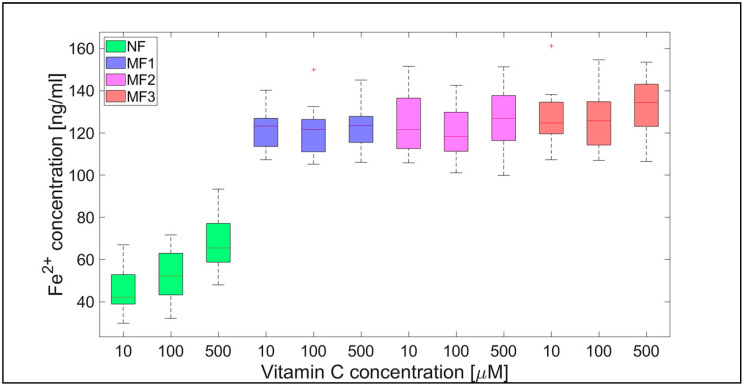
Box plot (with median value) of the released Fe^2+^ cations per hour from NF and MF with various LFs, induced by different concentrations of vitamin C (10, 100, and 500 µM) as a reducing agent over total experimental time (12 h). NF with LF = 868, MF1 with LF = 553, MF2 with LF = 733, and MF3 with LF = 873.

**Figure 5 ijms-21-06332-f005:**
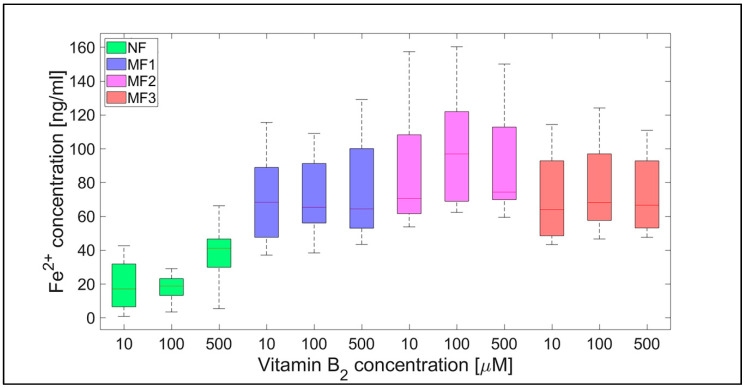
Box plot (with median value) of the released Fe^2+^ cations per hour from NF and MF with various LFs, induced by different concentrations of vitamin B_2_ (10, 100, and 1000 µM) as a reducing agent over total experimental time (12 h). NF with LF = 868, MF1 with LF = 553, MF2 with LF = 733, and MF3 with LF = 873.

**Figure 6 ijms-21-06332-f006:**
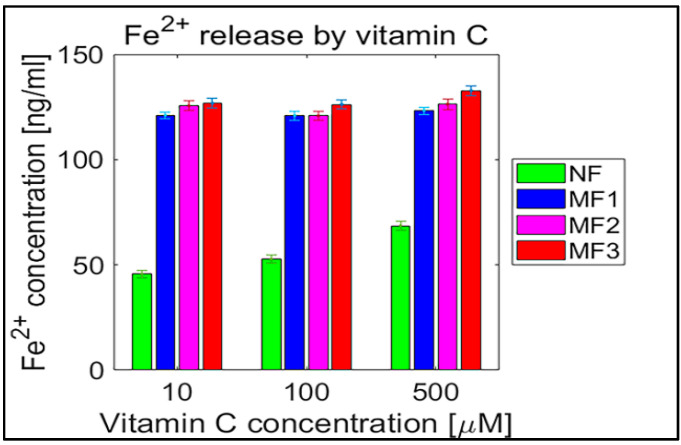
Confidence intervals of 95% (means ± 2 SEM), N = 120 of free ferrous ions after reduction from NF and MF mineral cores per hour induced by ascorbic acid (vitamin C). NF with LF = 868, MF1 with LF = 553, MF2 with LF = 733, and MF3 with LF = 873.

**Figure 7 ijms-21-06332-f007:**
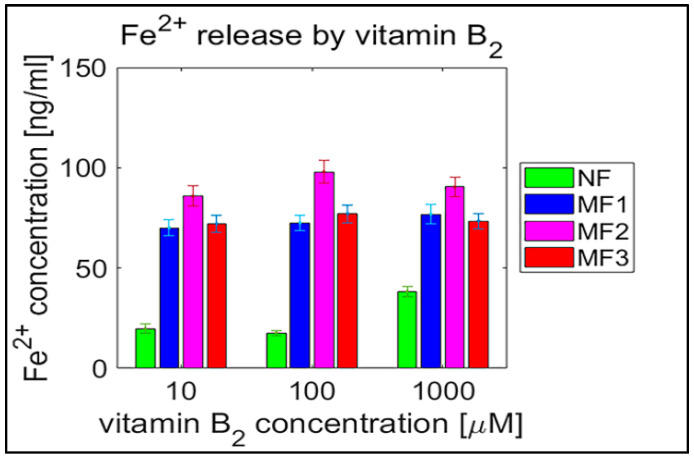
Confidence intervals of 95% (means ± 2 SEM), N = 120 of free ferrous ion concentration after reduction from NF and MF mineral cores per hour induced by riboflavin (vitamin B_2_). NF with LF = 868, MF1 with LF = 553, MF2 with LF = 733, and MF3 with LF = 873.

**Figure 8 ijms-21-06332-f008:**
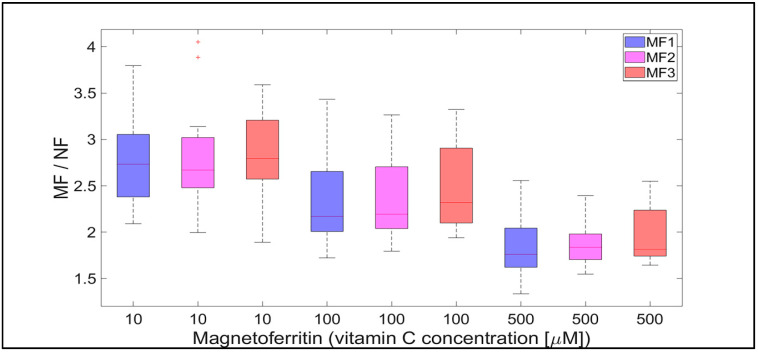
The ratio of MF with different LFs and NF per hour, induced by various concentrations of vitamin C (10, 100, and 500 µM) as a reducing agent over total experimental time (12 h). NF with LF = 868, MF1 with LF = 553, MF2 with LF = 733, and MF3 with LF = 873.

**Figure 9 ijms-21-06332-f009:**
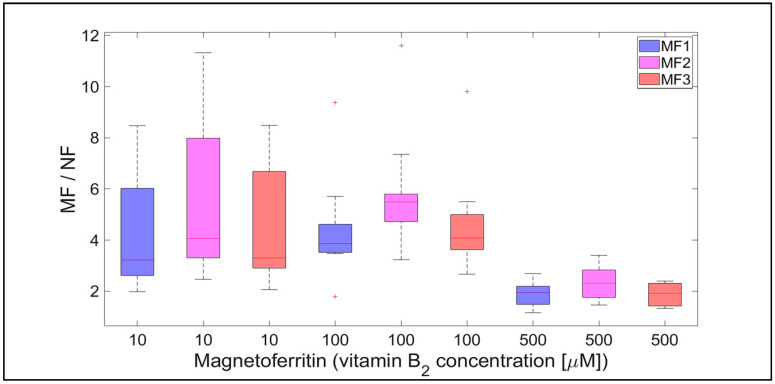
The ratio of MF with different LFs and NF per hour, induced by various concentrations of vitamin B_2_ (10, 100, and 500 µM) as a reducing agent over total experimental time (12 h). NF with LF = 868, MF1 with LF = 553, MF2 with LF = 733, and MF3 with LF = 873.

**Table 1 ijms-21-06332-t001:** The basic properties of prepared MF samples, including NF: LF, hydrodynamic diameter <D_HYDR_>, and zeta (ζ) potential.

Sample	LF	<D_HYDR_> [nm]	ζ Potential [mV]
MF1	553	46.00	−29.3
MF2	733	46.45	−26.9
MF3	873	42.20	−20.8
NF	868	15.60	−30.8

**Table 2 ijms-21-06332-t002:** Free ferrous ions concentration of reduced native ferritin and magnetoferritin.

Sample	Vitamin C [µM]			Vitamin B_2_ [µM]		
	10	100	500	10	100	1000
NF	45.7 ± 0.9	52.7 ± 1.0	68.6 ± 1.1	19.9 ± 1.2	17.7 ± 0.6	38.3 ± 1.3
MF1	121.0 ± 0.8	120.9 ± 1.1	123.3 ± 0.9	70.0 ± 2.1	72.4 ± 1.9	76.8 ± 2.4
MF2	125.6 ± 1.2	120.8 ± 1.0	126.3 ± 1.3	85.9 ± 2.6	98.0 ± 2.8	90.4 ± 2.4
MF3	126.7 ± 1.2	126.2 ± 1.2	132.7 ± 1.2	72.0 ± 2.1	77.0 ± 2.2	73.4 ± 1.9

Note: Confidence intervals of 95% (means ± 2 SEM), N = 120 of free ferrous ions after reduction from NF and MF mineral cores per hour, for both studied vitamins. Concentrations are in ng/mL. (NF—native ferritin with LF = 868, MF1—magnetoferritin with LF = 553, MF2—magnetoferritin with LF = 733, MF3—magnetoferritin with LF = 873.).

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
