# Peer review of "Quantification of Iron Release from Native Ferritin and Magnetoferritin Induced by Vitamins B2 and C"

_ijms, 2020, doi:10.3390/ijms21176332_

Round 1

Reviewer 1 Report

The paper entitled “Quantification of Iron Release from Native Ferritin and Magnetoferritin Induced by Vitamins B2 and C” is submitted for publication in International Journal of Molecular Sciences by Oliver Strbak and collaborators.

This research work consists in a spectroscopic assay of the release of free iron (II) ions from native ferritin and in the aim of comparison from magnetoferrotin as a model system. This study, is presented as the starting point of a full research project that aims to answer the fundamental question of the key role played by biogenic iron, when accumulated after reduction of iron from ferritin by natural reductant agent, ascorbic acid or vitamin B12 , in the development of various pathologies ,especially neuro deregenative diseases or cancers.

The first part of this paper describes the synthesis of three magnetoferritins varying by their loading factors, which have been analysed by UV-Vis spectroscopy. Their magnetic properties have been studied as well. Then, their interaction with the two vitamins are analysed via spectroscopic measurements and discussed. The iron release from native ferritin is clearly dependant of the concentration of the vitamins. Conversely, even if the amount of reduced iron is very higher for magnetoferritin, this release is not sensitive to the vitamins concentration. Accordingly, the accumulation of magnetite in the tissue with aging could produce, in the presence of vitamins C and B12, but not only, enough free radicals to induce oxidative stress and as results pathologies.

The whole manuscript, suitably written and well supported by an experimental section, this one completed with SI, would deserve to be published after minor corrections.

Page 2; the authors describe the structure of Ferritin and refer to the Figure 1 but the Figure 1 in the manuscript on page 4 displays the Magnetization curves of the three magnatoferritins which must be the figure 2.

Page 9 : in figure 9, the title of the axis of the abscissa is wrong, vitamin B12 concentration instead of vitamin C concentration

Author Response

Response to Reviewer #1

Reviewer #1: Page 2; the authors describe the structure of Ferritin and refer to the Figure 1 but the Figure 1 in the manuscript on page 4 displays the Magnetization curves of the three magnatoferritins which must be the figure 2.

Thank you for the comment. Indeed, Figure 1 in the older version of the manuscript did contain a picture of ferritin, but due to a large number of images in the results, we decided to omit this image, but forgot the link in the text.

Reviewer #1: Page 9 : in figure 9, the title of the axis of the abscissa is wrong, vitamin B12 concentration instead of vitamin C concentration.

Thank you, it was an inattention mistake. The title of the x-axis in Figure 9 has been corrected.

Reviewer 2 Report

Reference: 878429

Quantification of Iron Release from Native Ferritin 2 and Magnetoferritin Induced by Vitamins B2 and C

The manuscript describes the quantification and spectrometric release of ferrous ions from native ferritin and magnetoferritin in an in vitro model system and their reduction by vitamins C and B2. The strengths of the manuscript are the volume of evidence that showed a significant difference in the reduction and iron release from native ferritin compared to magnetoferritin for both vitamins. However, there are concerns with the style of presentation and the interpretation of the data. A criticism of the manuscript is the tendency towards verbosity and word overuse. In general, this could be improved by employing scientific terms to describe the processes and the concepts of the subject matter. Besides, first-person plural pronouns feature rather too often in the text.

For example, the underlined are not necessary, and this is common in the whole document.

‘Iron played an irreplaceable role in the origin of life on Earth [1], the consequences of which we can see today. Indeed, except for Lactobacillus and some strains of Bacillus [2], iron is an essential compound for living organisms from all three domains of life [3]. In present-day living systems, it is involved in many fundamental metabolic processes, including photosynthesis, respiration, DNA synthesis, cell proliferation, differentiation, etc. Its success very probably results from the wide range of redox potentials of iron-based complexes (from ≈ -500 to 600 mV)’.

a so-called

State the 3 domains of lfe? archea, pro & eukaryotes?

A list is presented below of suggestions on areas requiring the authors’ attention for an amendment to the text.

 ‘Iron is involved in many fundamental metabolic processes, including photosynthesis, respiration, DNA synthesis, cell proliferation and differentiation’ will be sufficient

Line 45 replace ‘them’ with ‘ toxic in systemic metabolism

47----delete ‘even’

Unfortunately, we still do not know whether iron accumulation is the initial cause or the consequence of the pathology.

Consider:

48-49--Although the preliminary view more or less points to iron as a source of toxicity, this issue remains open. ---delete the statement

53-54 ---- in their loss of functionality. The more free ferrous ions, the more free radicals, and the higher the oxidative stress that can result in an outbreak of disease [10].

What do you mean by loss of functionality? The whole sentence is too wordy and superficial

55-56--and also functional ferritin which is responsible for the management of the free iron

This is not strictly accurate or not well presented

Line 47-56 is a bit pedantic and it lacks the scientific description of the concepts at the level required for the manuscript. So also 69-81

67-68 --- It was proposed that a precursor of iron accumulation and mineralization is ferritin [18,19].  This seems like an incomplete sentence

82-83- The crucial point from the biomedical point of view is that the composition of the mineral core probably depends on the health of the subject.

This needs clarity & lacks substance

TABLE 2- write  brief legends and place what is presented at footnote beneath the Table, applicable to all the Tables

Line 276-296 in the Discussion is a repetition of part of the Introduction

351-352- Iron excretion occurs only at a basal rate regardless of iron absorption through uncontrolled processes,----underline factually incorrect, iron absorption is a regulated process

404-406 --- our study must be supplemented by the influence of other biochemical factors on the reduction and transformation of NF to pathological ferritin-containing magnetite.

Complemented by?

Major

The main issue with the manuscript is that the presentation needs modification to reflect the added value of the research and avoid speculations in the interpretation of the data. It seems that the authors are not specialists in iron metabolism research because some description of the concepts are superficial and a few factual inaccuracies are presented. Another concern of this reviewer is that the subject of biogenic magnetic nanoparticles is still evolving and in particular the physiological functions of magnetite and magnetoferritin require further investigations in vivo. Some areas that require clarity are listed below:

  1. Th authors stated that ‘protection against oxygen was facilitated by closing the Eppendorf tube’.

All reagents used in the analytical procedure must be free of oxygen in the presence of 2+Fe and 3+Fe to get accurate results. Available O2 will oxidize free 2+Fe and introduce errors in the analysis. Presumably all right since  3+Fe is supposedly chelated and 2+Fe was measured after its reduction by Vitamin C & A.

  1. Is it a fact that ferritin has not been shown by experimental evidence to participate in biomineralization of magnetite in vivo?
  2. ‘This confirmed accumulation of magnetite in the tissues with aging, (ageing) together with our results, however, indicate a new view on the toxic effect of iron and the possible emergence of pathological processes in tissue, including neurodegeneration’.

 The data presented were  in vitro analysis and caution is needed in the interpretation to in vivo conditions

  1. The amount of Fe2+ cations released from ferritin could indicate whether the transformed mineral core itself has the potential to initiate the disease, or if it is just a product of pathology.

Biomenralization is common in the brain & in different cancer types & certain diseases. Are increased levels of magnetite/ magnetoferritin  the cause or con­sequence of disease? The genes regulating the process unlike bacteria are not yet deciphered in humans

  1. In general, the writing seems too wordy and lacks the scientific style of presentation. Moreover, some sentences are oversimplified and concepts need to be presented scientifically.
  2. General recommendations
  • The manuscript is too long
  • Presence the evidence or write briefly on magnetoferritin in human tissues as well as the implications in physiology & pathology. The Introduction need be shortened to a third such that the style is more chemical than on iron metabolism.
  • The graphs, axes & legends need improvement to a higher quality.
  • The graphs in S4 & S5 seem better presented, Table 3 may not be necessary
  • Provide legends for all the figures, the quality of the graph is poor, the X-axis label can be better presented
  • Avoid too many personal pronouns
  • Rewrite the Discussion, include critical analysis, write briefly on the speculations, hypothesis and the limitations of the experiments, in vitro Vs in vivo, possible other sources of biogenic magnetic particles in human tissues, and they may not possibly be sources of labile free iron in human tissues.
  1. . Shorten the Discussion and analyse the data with literature evidence

And preserve ‘But to understand our reasons, we must make a small detour etc etc in a review article.----delete this part but mention briefly as suggested in the Discussion,

Author Response

Response to Reviewer #2

The manuscript describes the quantification and spectrometric release of ferrous ions from native ferritin and magnetoferritin in an in vitro model system and their reduction by vitamins C and B2. The strengths of the manuscript are the volume of evidence that showed a significant difference in the reduction and iron release from native ferritin compared to magnetoferritin for both vitamins. However, there are concerns with the style of presentation and the interpretation of the data. A criticism of the manuscript is the tendency towards verbosity and word overuse. In general, this could be improved by employing scientific terms to describe the processes and the concepts of the subject matter. Besides, first-person plural pronouns feature rather too often in the text.

Thank you for your feedback. Since we are all non-native English speakers, the manuscript before submission had undergone professional proofreading (copy-editing service) in ProofScience Ltd. (https://www.proofscience.com/), Dr. Jeremy Day as an editor. Nevertheless, we edited the text according to your recommendations and sent it for further proofreading.    

For example, the underlined are not necessary, and this is common in the whole document.

'Iron played an irreplaceable role in the origin of life on Earth [1], the consequences of which we can see today. Indeed, except for Lactobacillus and some strains of Bacillus [2], iron is an essential compound for living organisms from all three domains of life [3]. In present-day living systems, it is involved in many fundamental metabolic processes, including photosynthesis, respiration, DNA synthesis, cell proliferation, differentiation, etc. Its success very probably results from the wide range of redox potentials of iron-based complexes (from ≈ -500 to 600 mV)'.

Underline parts have been omitted.

a so-called

Deleted.

State the 3 domains of lfe? archea, pro & eukaryotes?

The following were meant: Bacteria, Archaea, and Eukarya. However, it has been omitted from the sentence. 

A list is presented below of suggestions on areas requiring the authors' attention for an amendment to the text.

 'Iron is involved in many fundamental metabolic processes, including photosynthesis, respiration, DNA synthesis, cell proliferation and differentiation' will be sufficient

Corrected.

Line 45 replace 'them' with 'toxic in systemic metabolism

Replaced.

47----delete 'even'

Deleted.

Unfortunately, we still do not know whether iron accumulation is the initial cause or the consequence of the pathology.

Consider:

48-49--Although the preliminary view more or less points to iron as a source of toxicity, this issue remains open. ---delete the statement

Deleted.

 53-54 ---- in their loss of functionality. The more free ferrous ions, the more free radicals, and the higher the oxidative stress that can result in an outbreak of disease [10].

"Loss" has been deleted. The following sentence has been rewritten: "The free ferrous ions produce more free radicals increasing the oxidative stress that can result in an outbreak of disease [10]."

What do you mean by loss of functionality? The whole sentence is too wordy and superficial

With the "loss of functionality" we mean impairment of the function of the protein, lipids and DNA due to oxidation by free radicals as is described in numerous publications, e.g. Phaniendra et al., 2015 and references within. This publication has been added to References (# 10). The word "loss" has been replaced by the word "impairing."

55-56--and also functional ferritin which is responsible for the management of the free iron

This is not strictly accurate or not well presented

The sentence has been modified as follows: "To prevent the oxidative damage, it is necessary to have a balanced ratio of free radicals and antioxidants to achieve equilibrium [12], and also functional ferritin which is crucial for the iron homeostasis [13]."

Line 47-56 is a bit pedantic and it lacks the scientific description of the concepts at the level required for the manuscript. So also 69-81

The lines 47-56 have been modified based on your recommendations (please see the answers above).

The lines 69-81 have been modified as follows: " In physiological conditions, the ferrous ions can quickly diffuse to the ferritin interior through the eight narrow hydrophilic three-fold channels [22]. During the transfer, Fe2+ cations are oxidised to ferric ions, forming an insoluble ferrihydrite-like mineral core in the ferritin's cavity. Oxidation of iron occurs in the H-subunit of the ferritin particle and the L-subunits assist iron nucleation and mineralisation [23]."

67-68 --- It was proposed that a precursor of iron accumulation and mineralisation is ferritin [18,19].  This seems like an incomplete sentence

The sentence has been extended: " Although we do not currently know the reason for the presence of magnetite in human tissue, it was proposed that a precursor of iron accumulation and magnetite mineralisation is ferritin [19,20]. "

82-83- The crucial point from the biomedical point of view is that the composition of the mineral core probably depends on the health of the subject.

This needs clarity & lacks substance

This sentence has been deleted. The modified version has been added to line 91: " From this point of view, the ferritin's mineral core composition could be associated with the presence of pathology. "

TABLE 2- write  brief legends and place what is presented at footnote beneath the Table, applicable to all the Tables

The legend has been added to the description of Tables 2 and 3: "(NF - native ferritin with LF = 868, MF1 - magnetoferritin with LF = 553, MF2 - magnetoferritin with LF = 733, MF3 - magnetoferritin with LF = 873.)" 

Line 276-296 in the Discussion is a repetition of part of the Introduction

The repetition part has been deleted and two introductory sentences have been added: " As previously described, the mineral core of physiological ferritin consists of ferrihydrite, while mineral core of ferritin associated with the presence of pathology consists of magnetite [24,25]. Moreover, the size of the magnetite nanocrystals is ranging from 1 to 70 nm [44]. "

351-352- Iron excretion occurs only at a basal rate regardless of iron absorption through uncontrolled processes,----underline factually incorrect, iron absorption is a regulated process

Yes, iron absorption is a regulated process, what we also state on the lines 351-352, including reference (# 46). What we say is that iron excretion is not a regulated process and no specific (regulatable) excretion mechanism has developed in humans, which can result in the accumulation of iron in human tissue with ageing, as has been shown in other living systems. For example in Mercadante et al., 2019 (added to References as # 48) is stated: "Iron excretion is believed to occur at a basal rate irrespective of intestinal epithelium, blood loss, and exfoliation of dead skin. " Please see also references no. 1 - 5 within this manuscript.

404-406 --- our study must be supplemented by the influence of other biochemical factors on the reduction and transformation of NF to pathological ferritin-containing magnetite.

Complemented by?

The sentence has been modified: " To better understand its role in tissue and to understand its role in the emergence of various pathological processes, requires further research, including the study of other biochemical compounds influence on the reduction and transformation of NF to pathological ferritin-containing magnetite."

Major

The main issue with the manuscript is that the presentation needs modification to reflect the added value of the research and avoid speculations in the interpretation of the data. It seems that the authors are not specialists in iron metabolism research because some description of the concepts are superficial and a few factual inaccuracies are presented. Another concern of this reviewer is that the subject of biogenic magnetic nanoparticles is still evolving and in particular the physiological functions of magnetite and magnetoferritin require further investigations in vivo. Some areas that require clarity are listed below:

Th authors stated that 'protection against oxygen was facilitated by closing the Eppendorf tube'.

All reagents used in the analytical procedure must be free of oxygen in the presence of 2+Fe and 3+Fe to get accurate results. Available O2 will oxidise free 2+Fe and introduce errors in the analysis. Presumably all right since  3+Fe is supposedly chelated and 2+Fe was measured after its reduction by Vitamin C & A.

When protecting the Eppendorf tube from oxygen penetration, there is no other option than to close the tube. With such a small volume of solution in the tube and taking into account the fact that the protein foams significantly when bubbling, we would lose the entire volume of solution that would spill out of the tube by immersing the needle with nitrogen and running it. Moreover, ferrosine - iron chelate, added to the reaction in excess before the addition of ferritin/magnetoferritin, when the reduction of iron started in the presence of vitamin C/B2, reacts immediately with the released Fe2+. The reaction is immediate, and a stable ferrozine-Fe(II) complex is formed. It is very unlikely that the trapped Fe2+ in the ferrozine structure would be oxidised by residual oxygen in solution.

Is it a fact that ferritin has not been shown by experimental evidence to participate in biomineralisation of magnetite in vivo?

Except publications mentioned in the manuscript [24,25], which provide in-vitro analysis of ferritin's mineral core, we have no knowledge about the direct in-vivo evidence of ferritin engagement in magnetite biomineralisation. However, based on this experimental evidence and a fact that magnetite needs specific conditions with the formation of the so-called germinal core to form the mineral and the supply of both Fe2+ and Fe3+ ions, it is very probable that ferritin is a precursor of all in-vivo magnetite biomineralisation in humans.   

'This confirmed accumulation of magnetite in the tissues with aging, (ageing) together with our results, however, indicate a new view on the toxic effect of iron and the possible emergence of pathological processes in tissue, including neurodegeneration'.

 The data presented were  in vitro analysis and caution is needed in the interpretation to in vivo conditions

Yes, we agree that caution is needed. Therefore we have made changes as follows:

"This confirmed accumulation of magnetite in the tissues with ageing, together with our results could indicate a new possible perspective on the toxic effect of iron and the possible emergence of pathological processes in tissue, including neurodegeneration. However, further in-vivo studies are needed to elucidate this effect."

The amount of Fe2+ cations released from ferritin could indicate whether the transformed mineral core itself has the potential to initiate the disease, or if it is just a product of pathology.

Biomenralization is common in the brain & in different cancer types & certain diseases. Are increased levels of magnetite/ magnetoferritin  the cause or con­sequence of disease? The genes regulating the process unlike bacteria are not yet deciphered in humans

This sentence should not indicate whether magnetite is the cause or the consequence of the disesase because we do not know the answer yet. It should have clarified the reason why we are doing these experiments. We accept that it may be confusing, so we have modified it as follows:

'A comparison of the amount of iron released from the cores of NF and MF as a model system of pathological ferritin could indicate whether a transformed magnetite-containing core has the potential to initiate oxidative stress (disease) or could be only a result of pathological processes."

In general, the writing seems too wordy and lacks the scientific style of presentation. Moreover, some sentences are oversimplified and concepts need to be presented scientifically.

The manuscript has been modified due to your recommendations.

General recommendations

The manuscript is too long

The manuscript has been shortened.

Presence the evidence or write briefly on magnetoferritin in human tissues as well as the implications in physiology & pathology. The Introduction need be shortened to a third such that the style is more chemical than on iron metabolism.

Magnetoferritin is a laboratory prepared artificial system that we use as a model system of pathological ferritin, because of ferritin envelope and magnetite core. As described in references no. 24,25, the pathological ferritin was found in humans with Alzheimer disease. Due to our knowledge, actually, there is no more information available about pathological ferritin. Introduction has been shortened with more emphasis on iron metabolism.

The graphs, axes & legends need improvement to a higher quality.

Figures have been improved.

The graphs in S4 & S5 seem better presented, Table 3 may not be necessary

Table 3 has been deleted.

Provide legends for all the figures, the quality of the graph is poor, the X-axis label can be better presented

Legends have been provided for all figures.

Avoid too many personal pronouns

The manuscript has undergone new professional proofreading. The number of personal pronouns has been reduced.

Rewrite the Discussion, include critical analysis, write briefly on the speculations, hypothesis and the limitations of the experiments, in vitro Vs in vivo, possible other sources of biogenic magnetic particles in human tissues, and they may not possibly be sources of labile free iron in human tissues.

. Shorten the Discussion and analyse the data with literature evidence

And preserve 'But to understand our reasons, we must make a small detour etc etc in a review article.----delete this part but mention briefly as suggested in the Discussion,

The Discussion has been rewritten according to your recommendation and comments.

Round 2

Reviewer 2 Report

The authors have addressed queries on the scientific content satisfactorily. However, there are still some issues with the style of presentation. 

Please see the example with the Abstract alone, the whole document require

such, hence the proofreading needs a bit more. 

Author Response

Dear Reviewer,

Thank you for your valuable comments. We have modified the manuscript according to your recommendations. The manuscript has again undergone the copy-editing process by Dr. Jeremy Day from ProofScience Ltd. However, because you marked your suggestions in a non-final version of the manuscript (clean_version), some modifications are not reflected in the manuscript with the highlighted changes (file: ijms-878429_revised2_(highlighted_changes)). Please focus on the final clean version of the manuscript (ijms-878429_revised2_(clean_copy)).

Sincerely yours
Oliver Strbak
